# Fecal Microbiota Perspective for Evaluation of Prebiotic Potential of Bamboo Hemicellulose Hydrolysate in Mice: A Preliminary Study

**DOI:** 10.3390/microorganisms9050888

**Published:** 2021-04-21

**Authors:** Nao Ikeyama, Mitsuo Sakamoto, Moriya Ohkuma, Shigeru Hiramoto, Jianpeng Wang, Shigenobu Tone, Kiwamu Shiiba

**Affiliations:** 1RIKEN BioResource Research Center, Microbe Division/Japan Collection of Microorganisms, Tsukuba, Ibaraki 305-0074, Japan; nao.ikeyama@riken.jp (N.I.); sakamoto@riken.jp (M.S.); mohkuma@riken.jp (M.O.); 2Division of Life Science and Engineering, Tokyo Denki University, Ishisaka, Hatoyama, Saitama 350-0394, Japan; hiramoto0531@gmail.com (S.H.); 19rmb05@ms.dendai.ac.jp (J.W.); tone@mail.dendai.ac.jp (S.T.)

**Keywords:** bamboo hemicellulose hydrolysates, fecal pH, serum cholesterol, short-chain fatty acids (SCFAs), fecal microbial composition

## Abstract

Bamboo hemicellulose hydrolysate (BHH) may possess antihypercholesterolemic activity; however, this activity requires further comprehensive study to assess the prebiotic mechanisms of BHH in vivo. Here, we used high-throughput 16S rRNA gene sequencing to preliminarily investigate the correlations between BHH and the fecal microbiomes of three groups of mice fed either a normal diet, a high-fat diet, or a high-fat diet supplemented with 5% BHH for 5 weeks. Alpha diversity (within community) was nonsignificant for all groups; however, beta diversity analysis among communities showed that 5% BHH suppressed the significant changes induced by the high-fat diet. The *Firmicutes*/*Bacteroidetes* ratio, the family S24-7 within the order *Bacteroidales*, the family *Lachnospiraceae* and several cellulolytic taxa were slightly ameliorated in the BHH group. These results indicated that BHH supplementation influenced the gut bacterial community and suppressed the high-fat diet-induced alterations. Additionally, BHH significantly lowered the serum cholesterol levels and fecal pH. Improving short-chain fatty acid production for all of the bacterial communities in the mouse guts may induce this effect. Thus, the prebiotic potential of BHH should be evaluated considering the gut microbial communities and their interactions.

## 1. Introduction

Bamboo is a common name for evergreen plants belonging to the subfamily *Bambooaceae*, which includes 1575 species and is found mainly in warm and humid climates in East Asia and South Asia [1]. It is one of the fastest-growing plants, and some species can elongate their culm section by more than 1 m per day during periods of maximum growth. It is a very useful biomaterial because it is a source of inexpensive reproducible biomass in these areas. The development of novel applications for bamboo is not only relevant to biomass utilization efficiency, but also the reduction of green gas emissions because bamboo has high carbon dioxide absorption capacities. Many studies on the health benefits of bamboo constituents have been conducted throughout the world since the 1960s, and it has been reported that extracts of various bamboo leaves, seeds, stems, and shoots have preventive effects on lifestyle-associated diseases such as oxidative stress, diabetes, obesity, and hyperlipidemia [2,3]. Li et al. investigated the potential prebiotic functions of bamboo shoot fiber and its alteration of the gut microbiome [4] and found that the different types of dietary fibers possessed individual activity; thus, the various fiber types should be studied to discover novel prebiotic fibers. Hemicellulose materials from bamboo extracts may confer the same health benefits as do bamboo shoots, which are immature bamboo. Additionally, dietary fibers in plant cell walls are important energy sources for the cecum and colonic microbiota. Specific anaerobic bacteria can metabolize complex carbohydrates via key enzymes and metabolic pathways, thus leading to the production of metabolites such as short-chain fatty acids (SCFAs) [5].

Acetate, propionate, and butyrate are SCFAs associated with regulating host metabolism, the immune system, and cell proliferation [6]. Additionally, oral intake of dietary fiber induces microbial diversity and SCFA production [7,8]. The gut microbiota composition, diversity, and richness are highly influenced by the diet, and SCFAs as bacterial metabolites from bamboo extracts may be important for prebiotic functions in the gut. Over the course of research on novel uses for bamboo, Maemura et al. have recently established a simple and cost-effective method for preparing bamboo hemicellulose hydrolysate (BHH) [9]. This method consists of high-temperature loading followed by enzymatic treatment [9]. They found that bamboo hemicellulose hydrolysate (BHH) has two unique properties. One property is its strong antioxidative activity in vitro. BHH can scavenge DPPH radicals with an intensity equivalent to 2.01 μmol of ascorbic acid per mg. The second property is a suppressive effect against plasma cholesterol elevation, accompanied by increased SCFAs in feces in high-fat diet-fed mice (in vivo). These results suggest that BHH may have beneficial functions in vivo via bioactive metabolites, such as SCFAs produced by gut bacteria, for maintaining host health.

Here, we preliminarily investigated the bacterial communities in mouse feces via 16S rRNA gene sequencing to determine whether administering 5% BHH would alter the gut microbiome and lower cholesterol in high-fat diet-fed mice.

We aimed to determine the alterations and interactions between BHH and the gut microbiome via high-throughput sequencing of the 16S rRNA genes. These results may enable a better understanding of the prebiotic potential of BHH mediated by bacterial metabolites.

## 2. Materials and Methods

### 2.1. Animals and Experimental Design

Nine-week-old male C57BL/6J mice were purchased from CLEA Japan. The mice, housed in plastic cages, were placed in an environment maintained at 23 ± 1 °C and 55 ± 5% relative humidity under a 12 h/12 h light/dark cycle.

The experimental protocol is shown in Figure 1. Mice were acclimatized for 1 week while being fed CLEA Rodent Diet CE-2 (CLEA Japan) and given Milli-Q water. All mice were kept separately, with free access to water and food. After 1 week of acclimatization, mice were divided into 3 groups according to body weight. The normal group (*n* = 3) was given a normal CE-2 diet, the control group (*n* = 4) was given high-fat diet 32 (HFD-32; CLEA Japan), the 5% BHH group (*n* = 4) was given the diet consisting of 95% HFD-32 and 5% BHH for 5 weeks. In a previous paper, the modified method and the chemical contents of BHH were reported [9]. The normal diet (CLEA Rodent Diet CE-2) was composed of 9.0% moisture, 24.8% crude protein, 4.6% crude fat, 4.7% crude fiber, 7.0% crude ash, and 49.9% nitrogen-free extract (NFE), with 340.2 Kcal/100 g of total energy. The high-fat diet (CLEA HFD-32) was composed of 6.2% moisture, 25.5% crude protein, 32.0% crude fat, 2.9% crude fiber, 4.0% crude ash, and 29.4% nitrogen-free extract (NFE), with 507.6 Kcal/100 g of total energy. The dietary consumption and body weight of each mouse were measured every week, respectively. All mice were kept separately, with free access to water and food during the experimental period. On the final day of the feeding period, feces from each mouse were collected immediately after defecation, and then all mice were withheld food for 16 h. The blood of each mouse was collected from the abdominal vein, and the serum was prepared. The feces and the sera were stored at −20 °C until analysis.

All experimental procedures were performed in accordance with institutional guidelines for animal research, and the study was approved by the animal ethics committee of Tokyo Denki University.

### 2.2. Chemical Analysis

To reconfirm the physiological changes by feeds of BHH and high-fat diets, such as plasma cholesterol and fecal pH in the previous study [9], the chemical analysis was performed. The concentrations of total cholesterol (T-Cho) and triglyceride (TG) in the serum were determined using Cholesterol E-test Wako kit (Fuji Film) and Triglyceride E-test Wako kit (Fuji Film), respectively. Lipid peroxides in the serum were expressed in terms of malondialdehyde (MDA) equivalents, which were measured as thiobarbituric acid-reactive substances (TBARS) by a TBARS Assay Kit (Cayman Chemical, Ann Arbor, MI, USA). The fecal pH was determined according to the method used by Maemura et al. [9].

### 2.3. DNA Extraction from Feces

A fecal sample (50 mg) from each mouse was weighed and transferred to a ZR BashingBead™ Lysis Tube (ZYMO RESEARCH, Irvine, CA, USA), then dissolved in 750 µL of BashingBead™ Buffer (ZYMO RESEARCH) and homogenized for 5 min using Vortex-Genie 2 (Scientific Industries, Bohemia, NY, USA). DNA was extracted from the homogenized feces using a Quick-DNA™ Fecal/Soil Microbe Miniprep Kit (ZYMO RESEARCH) according to the manufacturer’s instructions. The DNA concentrations of each sample were determined using the Qubit High-Sensitivity Assay and the Qubit 4 Fluorometer (Thermo Fisher Scientific, Waltham, MA, USA).

### 2.4. Microbiota Analysis by 16S rRNA Gene Sequencing

Briefly, the bacterial compositions were assessed by high-throughput sequencing of the 16S rRNA gene following 16S metagenomic sequencing library preparation using the Nextera XT DNA Library Preparation Kit by Illumina. The first PCR step for amplification of microbial DNA (5 ng/µL) using 16S Amplicon PCR Forward Primer (5′ TCGTCGGCAGCGTCAGATGTGTATAAGAGACAGCCTACGGGNGGCWGCAG) and 16S Amplicon PCR Reverse Primer (GTCTCGTGGGCTCGGAGATGTGTATAAGAGACAGGACTACHVGGGTATCTAATCC) targeting the 16S V3-V4 region was performed using the following program: 95 °C for 3 min followed by 25 cycles of 95 °C for 30 s, 55 °C for 30 s, 72 °C for 30 s, and a final elongation step at 72 °C for 5 min. The second PCR was performed to add the adapter sequences using the following run conditions: 95 °C for 3 min, followed by 8 cycles of 95 °C for 30 s, 55 °C for 30 s, and 72 °C for 30 s, and a final elongation step at 72 °C for 5 min. For each PCR clean-up step, AMPure XP beads were used to remove free primers and primer-dimer species, and the PCR products were quantified using the QuantStudio 3D Digital PCR system (Thermo Fisher Scientific). The final purified amplicons were pooled together (normalized to 4 nM) and sequenced on a MiSeq using the 3v-600 cycle kit with 2 × 300 bp paired ends.

The QIIME2 (wrapper package Quantitative Insights Into Microbial Ecology) pipeline version 2019.10 [10] was used at default parameters for all analyses of the microbial community. The DADA2, wrapped in QIIME2, was used for denoising and quality control of paired-end sequences. The processed fastq files determining the operational taxonomic units (OTUs) were profiled against the Greengenes database [11].

Alpha- and beta-diversity analyses were performed at the lowest number of sequences rarefied to avoid bias by sampling depth. Taxonomic comparisons for each treatment were analyzed by alpha diversity (within community) based on the observed OTUs, Shannon’s diversity index, and Pielou’s measure of species evenness. Beta diversity (among communities) was analyzed using permutational multivariate analysis of variance (PERMANOVA). Principal coordinate analyses (PCoAs) were performed based on Bray–Curtis distances and unweighted and weighted UniFrac distances and visualized using three-dimensional (3D) plots in EMPeror [12].

### 2.5. Statistical Analysis

All data were analyzed using one-way analyses of variance with Excel statcel3 software, followed by the Dunnet’s test for the comparison between the test and control groups or the Tukey–Kramer test for individual differences between groups. The statistical significances of alpha- and beta-diversity were calculated by Kruskal–Wallis test. All data were expressed as mean ± SD, *p*-values < 0.05 were considered to indicate significant differences.

## 3. Results

### 3.1. Effects of BHH on Body Weight, Food Intake, and Water Intake

Table 1 shows the body weights, food intakes, and water intakes of the three groups during the experimental period. The calorie-based food intakes, the weight gains, and final body weights of the high-fat diet groups (the control and the 5% BHH group) were significantly higher than those of the normal diet group. However, these significances between the control and the 5% BHH group were not observed (*p* > 0.05). These results were similar to current published reports [9].

### 3.2. Effects of BHH on Serum Total Cholesterol, Triglycerides, Malondialdehydes (MDA), and Fecal pH

Table 2 shows total cholesterol, triglycerides, and MDA levels in the serum collected from mice 16 h after the end of the diet treatments. The normal group had the lowest cholesterol levels, followed by the 5% BHH group and the control group (all three groups were significantly different). There was no significant difference in triglyceride levels between the three groups. For MDA levels which is a representative biomarker of oxidative stress in the serum [13], the control and the 5% BHH group were significantly higher than the normal group, but there was no significant difference between the control and the 5% BHH group.

The pH of the feces of the control group collected on the final day of the intake period was significantly higher than that of the normal and the 5% BHH group. There was no significant difference in pH between the normal and the 5% BHH group (Figure 2A). The pH level correlated positively (*r* = 0.823, *p* < 0.01) with the serum total cholesterol concentration (Figure 2B). As expected, these results were in agreement with a previous study [9].

### 3.3. Effect of BHH on Fecal Microbiota Composition

In this study, 11 mouse fecal samples (3 from the normal group, 4 from the control group, and 4 from the 5% BHH group) were used, however, one of the 5% BHH samples was excluded from the bioinformatic analysis due to it having too few reads.

The number of sequences in the normal group were 171,462 ± 5950, the control group were 197,961 ± 27,108, the BHH group were 202,813 ± 52,408, respectively.

Alpha-diversity based on the observed OTUs, Shannon diversity index, and Pielou’s evenness measure in all groups showed *p* = 0.588, 0.705, and 0.554 by a Kruskal–Wallis test, respectively. The box plots are shown in Figure 3. The observed OTUs in the normal group were 219 ± 19, the control group were 240 ± 36, the BHH group were 259 ± 69. In addition, the pairwise statistical significances of normal vs. control (*p* = 1.00 (observed OTUs), 0.480 (Shannon diversity index) and 0.289 (Pielou’s evenness measure)), normal vs. 5% BHH (*p* = 0.513 (observed OTUs), 0.513 (Shannon diversity index), and 0.513 (Pielou’s evenness measure)), and control vs. 5% BHH (*p* = 0.289 (observed OTUs), 0.724 (Shannon diversity index) and 0.724 (Pielou’s evenness measure)) indicated that there was no significant difference in the alpha-diversities associated with each treatment.

Beta-diversities for each treatment were analyzed using permutational multivariate analyses of variance (PERMANOVA) with 999 permutations. Bray–Curtis distance showed *p* = 0.003 (in all), *p* = 0.033 (normal vs. control groups), *p* = 0.126 (normal vs. 5% BHH groups), and *p* = 0.036 (control vs. 5% BHH groups). Unweighted UniFrac distance showed *p* = 0.003 (in all), *p* = 0.032 (normal vs. control groups), *p* = 0.501 (normal vs. 5% BHH groups), and *p* = 0.026 (control vs. 5% BHH groups). Weighted UniFrac distances showed *p* = 0.001 (in all), *p* = 0.031 (normal vs. control groups), *p* = 0.092 (normal vs. 5% BHH groups), and *p* = 0.020 (control vs. 5% BHH groups). Consequently, although there were significant differences in beta-diversity when comparing the control vs. normal groups and the control vs. 5% BHH groups, the beta-diversity of the normal and 5% BHH groups was insignificant. These results indicated that microbial diversity in the BHH group was similar to that of the normal group and differed significantly from that of the control group. Figure 4 shows the 3D PCoA plot based on the Bray–Curtis distance and unweighted and weighted UniFrac phylogenetic distances.

Subsequently, taxonomic comparisons were performed. Two taxa (*Bacteroidetes* and *Firmicutes*) made up the majority of the total bacterial reads at the phylum level (Figure 5A). Previous studies revealed that the ratio of *Firmicutes* to *Bacteroidetes* (the F/B ratio) was increased by a high-fat diet [14,15]. In our study, although not statistically significant, the F/B ratio was increased from 0.645 on the normal diet to 1.003 on the control diet and the value remained at 0.862 in the BHH group (Figure 5B).

At the family level, 39.99 ± 9.03% of family S24-7, within the order *Bacteroidales*, was the most abundant in the normal group, followed by *Lachnospiraceae* (22.14 ± 12.46%), *Bacteroidaceae* (9.38 ± 3.26%), and *Prevotellaceae* (5.08 ± 1.67%) (Figure 5C). However, in the high-fat diet group (the control group), family S24-7 (4.75 ± 0.96%), *Lachnospiraceae* (12.57 ± 3.89%), and *Prevotellaceae* (0.12 ± 0.04%) were decreased (Figure 6). Family S24-7 is frequently detected by metagenomic analyses of animal fecal samples, including feces from mice [16]. The decrease in family S24-7 was slightly suppressed by 5% BHH (11.01 ± 2.02%). Moreover, the relative abundance of *Lachnospiraceae* in 5% BHH (21.92 ± 4.37%) was similar to the abundance in the normal group (22.14 ± 12.46%) (Figure 6). The relative abundance of *Ruminococcaceae* was 4.75 ± 2.40% in the normal group, and it increased to 11.57 ± 4.01% on the high-fat diet. Notably, the family *Ruminococcaceae* was detected at 14.47 ± 7.19% in the 5% BHH group. *Lactobacillaceae* was highly presented in the control group (9.18 ± 5.47%), however, significantly lower detection (0.08 ± 0.05%) was observed in the 5% BHH group. The relative abundance of *Prevotellaceae* showed a significant decrease from 5.08 ± 1.67% to 0.12 ± 0.04% following the control diet, and a relative abundance of 0.25 ± 0.10% was observed in the 5% BHH group. The relative abundance of *Bacteroidaceae* was not different among all groups.

Only 20.88–29.55% of the total bacterial content was identified below the genus level. The bacteria with the highest relative abundances at this level were: in the normal group, *Bacteroides acidifaciens* (9.38 ± 3.26%), *Prevotella* (5.00 ± 1.08%), *Lactobacillus* (2.46 ± 0.25%), *Clostridium* within *Lachnospiraceae* (1.88 ± 1.57%), and *Clostridium disporicum* (3.34%); in the control group, *Bacteroides acidifaciens* (9.70 ± 5.10%), *Lactobacillus* (4.11 ± 2.08%), *Clostridium* within *Lachnospiraceae* (3.87 ± 0.97%), *Clostridium disporicum* (2.10 ± 1.42%), *Dorea longicatena* (1.96 ± 0.64%), *Clostridium ruminantium* (1.28 ± 0.98%), and *Marvinbryantia formatexigens* (1.02 ± 0.24%); and in the 5% BHH group, *Bacteroides acidifaciens* (7.81 ± 1.85%), *Clostridium* within *Lachnospiraceae* (2.39 ± 1.51%), *Marvinbryantia formatexigens* (2.06 ± 1.26%), *Sporobacter termitidis* (1.79 ± 0.80%), and *Dorea longicatena* (1.03 ± 0.16%). In the BHH group, the relative abundance of *Sporobacter termitidis* was significantly higher in all groups. Although these taxonomic comparisons were not dramatically altered, BHH and the high-fat diet slightly shifted the microbial diversity.

## 4. Discussion

Maemura et al. showed that increased plasma cholesterol levels were significantly ameliorated in mice fed a high-fat diet supplemented with 5% BHH compared with those of the control mice fed a high-fat diet alone [9]. Based on our previous study, we preliminarily investigated the bacterial communities in mouse feces via 16S rRNA gene sequencing to determine whether administering 5% BHH would alter the gut microbiome in addition to exerting other physiological changes.

Hemicellulose components help support health, control high blood pressure, and decrease serum LDL (low-density lipoprotein) cholesterol concentrations [17,18,19]. Maemura et al. showed an efficient method for preparing hemicellulose oligosaccharides from bamboo using a high-temperature treatment followed by an enzymatic reaction using cell-wall lytic enzymes [9]. Furthermore, continuous intake of high-fat diets causes vascular disorder, obesity, and dysbiosis in the intestinal microbiota. Thus, we assessed the suppressive effects of administering 5% BHH to mice and compared them with the effects of feeding high-fat and normal diets.

Alpha diversity was analyzed to compare the gut microbiome within each group based on species richness and evenness (observed OTUs, Shannon’s diversity index, and Pielou’s measure of species evenness). Neither the high-fat diet nor 5% BHH administration statistically significantly altered the alpha diversity. However, beta-diversity analysis (species diversity among groups) showed statistically significant differences between the normal and control and between the control and 5% BHH groups (*p* < 0.05). Notably, the 5% BHH and normal groups exhibited no significant difference in beta diversity. A recent study showed that westernized diets are associated with a loss of species diversity in the gut microbiome [20], and cellulose as insoluble fiber significantly altered the microbial community compositions more than did soluble fibers such as short-chain fructooligosaccharides and pectin [21]. These results indicated that BHH supplementation influenced the gut bacterial community and suppressed the alterations associated with high-fat diets.

Although the F/B ratios did not significantly differ between the groups, the ratio in the BHH group (0.862 ± 0.25) was recovered slightly to that of the normal group (0.645 ± 0.19) compared with that of the control group (1.003 ± 0.16). This indicated that the bacterial composition of the BHH group shifted taxonomically to that of the normal group.

The high-fat diet significantly decreased the relative abundance of the family S24-7 within the order *Bacteroidales* and family *Lachnospiraceae*, which produces SCFAs. The 5% BHH slightly suppressed these decreases (not significant). Despite the widespread occurrence of family S24-7 in the animal gut, which has been reported previously, the physiological function of these bacteria is still unclear [16]. However, a genomic analysis of family S24-7 revealed their ability to degrade complex plant cell wall glycans, such as hemicellulose and pectin [22]. This unrevealed organism, family S24-7, may play an important role in the biological activity of the gut.

*Ruminococcaceae* was the most highly detected in the BHH group than another group, but not significantly. *Ruminococcus*, the representative genus in the family *Ruminococcaceae*, degrades cellulose [23]. Among the less populated bacteria (<3%), several abundant taxa were increased in the BHH group. *Marvinbryantia formatexigens*, of the family Lachnospiraceae, and *Sporobacter termitidis*, isolated from wood-feeding termites [24,25,26] were increased via BHH. *M. formatexigens*, as illegitimate homotypic synonym of *Bryantella formatexigens* Wolin et al., 2004, is cellulolytic and produces succinate, lactate, and acetate via glucose fermentation with low concentrations of formate [24,25]. *S. termitidis* can use ferulate as energy and can produce acetate from 3,4,5-trimethoxycinnamate in the presence of sulfide or cysteine.

Interestingly, these analyses showed that the enrichments of *Bacteroidaceae*, *Prevotellaceae*, and *Lactobacillaceae*, which produce SCFAs were not significant or decreased in the BHH group, and *Lactobacillaceae* was most abundant in the control group. In a previous study, a high-fat diet increased the abundance of *Lactobacillaceae* and decreased the abundance of S24-7 in mice [27]. Some *Lactobacillaceae* are representative probiotic bacteria against metabolic and immune disorders [28]. A previous study using real-time PCR on microbial communities showed that *Lactobacillus* spp. were detected in significantly higher concentrations in obese patients than in lean controls or anorexic patients [29]. In the present study, the high-fat diet-fed mice showed significant increases in body weight and weight gain, but BHH administration did not suppress these physiological changes. This indicated that no association existed between BHH and obesity. However, the relative abundance of *Lactobacillaceae* in the BHH group was significantly decreased, and further long-term treatment may reveal a relationship between BHH, obesity, and *Lactobacillus* spp. Additionally, this result may have been caused by individual and specimen differences [30].

Thus, as a preliminary study, although the significant changes in specific bacteria via BHH administration remain unclear, the taxonomic composition may have been ameliorated because the BHH altered the bacterial diversity and relative abundances of several taxa.

Many studies have observed that bamboo extracts containing polysaccharides can reduce cardiovascular disease risk factors, such as hypercholesterolemia and oxidative stress [31,32,33,34,35]. A higher total cholesterol concentration (T-Cho) was detected in the control group serum than in the serum of the group fed a normal diet. Notably, the T-Cho was significantly decreased in the 5% BHH group. Maemura et al. (2016) obtained similar results [9]. Thus, BHH may possess resistance to hypercholesterolemia. However, as mice are HDL (high-density lipoprotein) dominant animals, the cholesterol component was not determined in this study. It should be considered that the differences in lipid metabolism between mice and humans in further studies. No effect on serum triglycerides was observed in the BHH group. In addition, although it was reported that the antioxidative activity of BHH in vivo was evaluated by MDA, a representative oxidative stress biomarker in the serum [13], in high-fat diet-fed mice previously, our results did not show significant suppression of the increase of MDA following administration of BHH as a 5% mixture with feed. Zhao et al. found that orally administered feruloyl-arabinoxylan hardly entered the bloodstream in rats, and this is consistent with our present results [36]. This suggests that the antioxidative compounds in BHH, which are estimated to be ferulate esters of oligosaccharides, are not absorbed in the intestine. However, because orally administered nondigestible materials are metabolized by gut bacteria [7], their bioavailability and prebiotic consumption could selectively depend on the microbial composition in the gut.

In the present animal experiments (see Materials and methods), the fecal pH, as well as the serum cholesterol level, decreased in the 5% BHH group compared with the control group, and a positive correlation was observed between the serum cholesterol level and the fecal pH of all mice (*r* = 0.823, *n* = 11). The lower fecal pH caused by 5% BHH indicates the alteration of gut environment and bacterial metabolism.

The specific anaerobes, *Lachnospiraceae*, *Bacteroidaceae*, and *Prevotellaceae*, could be associated with the presence of SCFAs and a decreased fecal pH. The p*K*a of most SCFAs is 4.6–4.9 (acetate is 4.76, butyrate is 4.82, propionate is 4.87). A previous study revealed the amount of fecal organic acid in mice fed a high-fat diet with BHH [9]. Maemura et al. detected high rates of acetate, propionate, and butyrate, which significantly differed from those of only the high-fat diet-fed group [9]. Propionate was the most highly abundant in the feces of the BHH group (811 ± 55.5 µg/g). SCFAs are bacterial metabolites from cellulose and hemicellulose produced via anaerobic fermentation and could be energy sources in the gut. Furthermore, the acidic conditions in the gut and conversion of cellulose into fermentable carbohydrates may improve butyrate production [37,38]. Thus, cellulose degradation mediated stimulation of bacterial growth and metabolism following an increase in fermentable saccharides. However, 16S rRNA gene sequencing showed that BHH did not significantly alter the major relative abundances of the bacterial taxa that comprised the mouse gut microbiome. In our previous study, we used physiological and transcriptomic analysis to study the microbial interaction via bacterial metabolites and reported that *Phascolarctobacterium faecium* was a succinate-using bacterium, and *Bacteroides thetaiotaomicron* was a saccharolytic bacterium that supported concomitant propionate production via the succinate pathway [39]. Although monocultured *P. faecium* is asaccharolytic and produces no organic acids as end products, a large amount of propionate and acetate were detected in the co-culture with *B. thetaiotaomicron*. These bacterial interactions could benefit host health. These results suggest that several factors, such as the concentrations of vitamins, SCFAs, and amino acids and the regulation of bacterial growth, metabolism, and pH, were possibly altered by BHH in the gut microbial community. Consequently, BHH may support interactions in SCFA production between asaccharolytic bacteria and cellulose-degrading bacteria.

SCFA production as bacterial metabolites may benefit lipid metabolism, intestinal immunity, and barrier function in the host [8,40]. Thus, BHH is fermented by the bacterial community in the intestinal tract to produce SCFAs, which help lower blood cholesterol mainly by decreasing the cholesterol synthesis rate in the liver [41].

Although we showed one effect of BHH on the gut bacterial composition in mice, this preliminary study had several limitations. First, this study had a small sample size (*n* = 3 or 4) for determining whether BHH possesses prebiotic effects. Further studies should use larger samples for statistical analysis. Second, long-term BHH administration is required. No interaction between BHH supplementation and weight gain was evidenced here. Continuous administration may provide better results. Third, high throughput 16S rRNA gene sequencing could have induced bias due to the number of PCR steps, the target marker gene region, and low detection sensitivity for taxonomic definitions below the genus level [42]. Several PCR artifacts should be considered to taxonomically analyze the gut microbiome via 16S rRNA genes. Furthermore, whole-genome sequencing would help determine the internal interactions by gene function. Fourth, we used only fecal samples from mice for microbial analysis. The microbiome profiles varied owing to the different sample locations such as the gut, mucosal tissue, and stool [30].

In summary, BHH altered the physiological responses, such as a decreased T-Cho and fecal pH, and the bacterial diversity and populations of several taxa in the mouse feces. Our study was the first to focus on how administering BHH affects the intestinal microbiome. These phenotypic outcomes were related to the compositions of the resident microbiota and could help clarify the prebiotic functions of BHH in future studies. However, which bacteria are involved in SCFA production from BHH in the intestinal tract remains unknown. These results suggest that the entire bacterial community and interactions in the mouse guts improved SCFA production and helped lower cholesterol. Further molecular biological and taxonomic analysis is required to specify the key bacteria in BHH-fed mouse intestines. Microbiological interactions between dietary factors and gut microbes must be considered to assess prebiotic activities. The chemical composition and physicochemical properties of different dietary fibers may be selectively linked to alterations in the gut microbiome. This study enables further understanding of novel materials with prebiotic potentials, such as BHH.

## Figures and Tables

**Figure 1 microorganisms-09-00888-f001:**
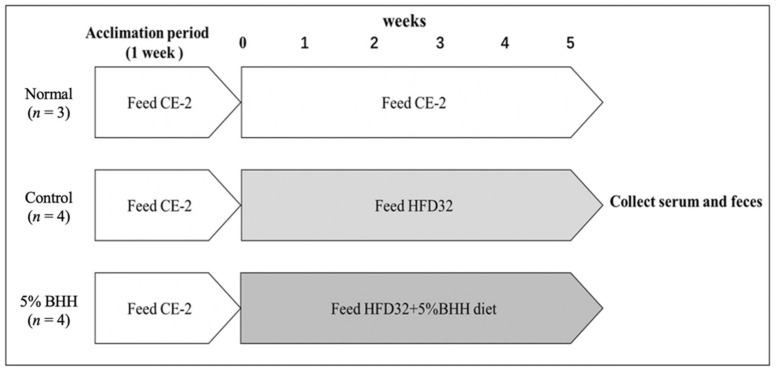
Experimental design. Eleven male C57BL/6JJcl mice (CLEA Japan Tokyo) of 9 weeks were used and the experiments were conducted after an acclimation period of 1 week. Normal, Normal group (*n* = 3); Control, Control group (*n* = 4); 5% BHH, Bamboo hemicellulose hydrolysate group (*n* = 4).

**Figure 2 microorganisms-09-00888-f002:**
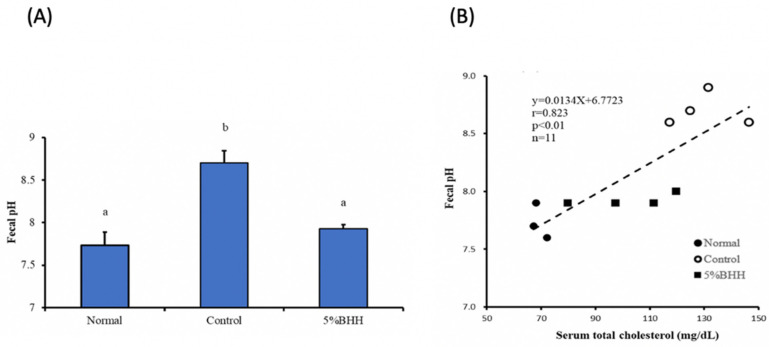
Effect of BHH on fecal pH (**A**) and the relationship between serum cholesterol and fecal pH (**B**). Bars in (**A**) are the average ± SD (*n* = 3 (Normal), *n* = 4 (Control and 5% BHH)). Different letters above the bars indicate significant differences.

**Figure 3 microorganisms-09-00888-f003:**
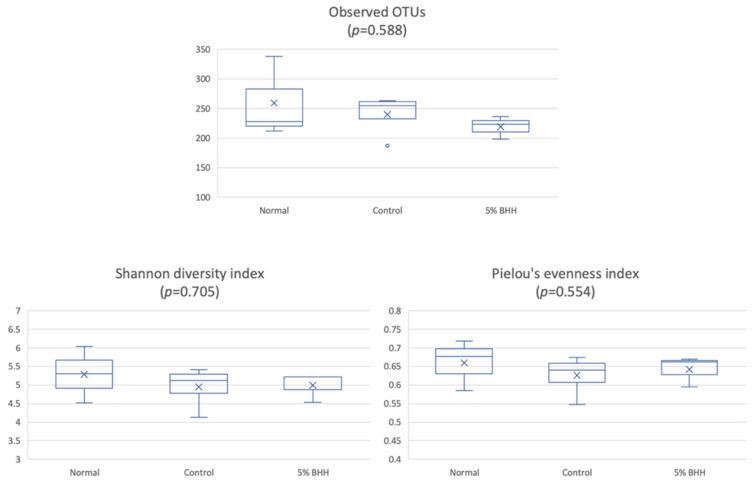
Alpha-diversity among each treatment group, Normal, Control, and 5% BHH by the observed OTUs, Shannon diversity index, and Pielou’s evenness index. The *p*-value by Kruskal–Wallis test in all groups. Normal (*n* = 3), Control (*n* = 4) and 5% BHH (*n* = 3) are used.

**Figure 4 microorganisms-09-00888-f004:**
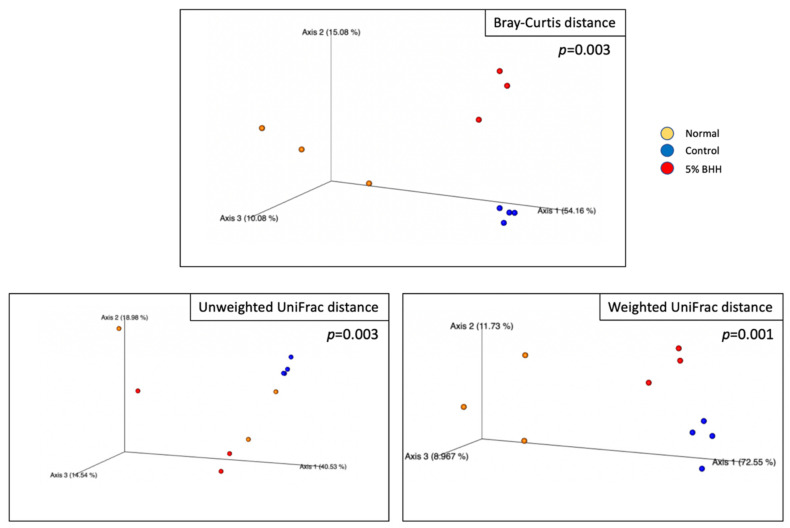
Visualization of PCoA 3D plots based on Bray–Curtis distance, unweighted UniFrac distance, and weighted UniFrac distance by EMPeror. The *p*-value by Kruskal–Wallis test in all groups. Yellow marker, Normal; blue marker, Control; red marker, 5% BHH.

**Figure 5 microorganisms-09-00888-f005:**
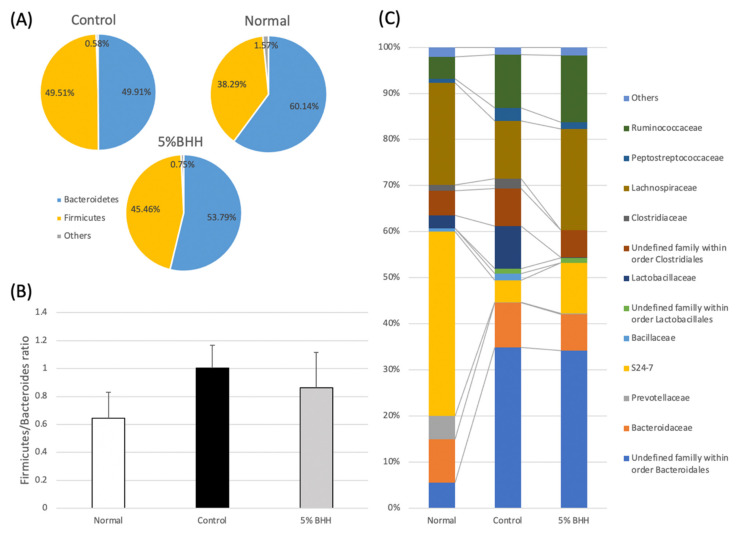
Differential bacterial community on the phylum-level (**A**), *Firmicutes*/*Bacteroidetes* ratio (**B**), and differential bacterial community on the family-level (**C**) among each treatment group, Normal, Control, and 5% BHH. Error bars represent the mean ± standard deviation. Others in (**A**) contained *Actinobacteria*, *Proteobacteria*, *Spirochaetes*, *Tenericutes*, and undefined bacteria on the kingdom level.

**Figure 6 microorganisms-09-00888-f006:**
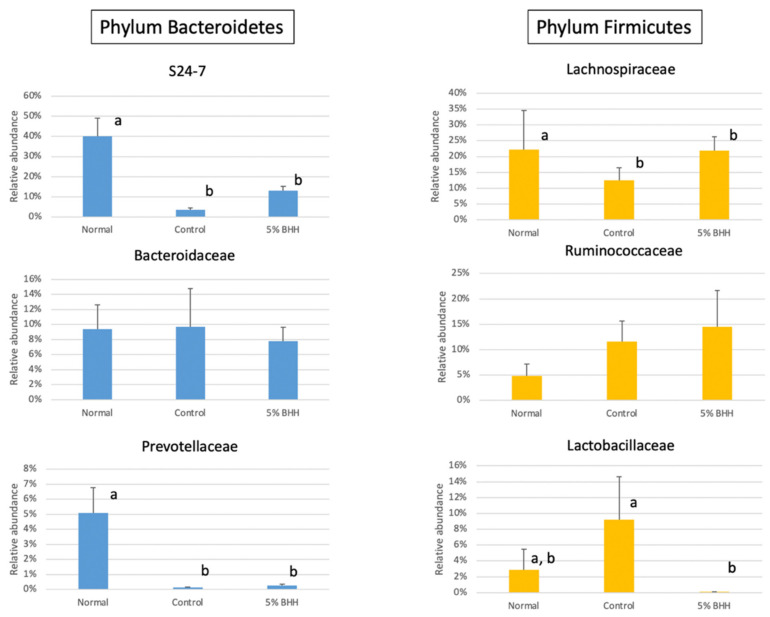
The differential relative abundance of taxa within the *Bacteroidetes* and *Firmicutes* phylum among each treatment group, Normal, Control, and 5% BHH. Error bars represent the mean ± standard deviation. Different letters above the bars indicate significant differences.

**Table 1 microorganisms-09-00888-t001:** Weight gain, final weight, food intake, and water intake in mice fed normal diets (Normal), high-fat diets (Control), or 5% BHH-containing high-fat diets (5% BHH) for 5 weeks.

Diets	Final Body Weight (g)	Weight Gain (g/5 weeks)	Food Intake		Water Intake (mL/5 weeks)
			(g/5 weeks)	(kcal/5 weeks)	
Normal	24.98 ± 1.40 ^a^	1.75 ± 1.35 ^a^	104.35 ± 2.25	355.26 ± 7.66 ^a^	266.0 ± 18.0
Control	30.95 ± 1.69 ^b^	7.60 ± 2.35 ^b^	91.30 ± 11.8	463.43 ± 59.89 ^b^	240.0 ± 12.0
5% BHH	29.87 ± 2.31 ^b^	6.75 ± 1.35 ^b^	89.92 ± 15.85	442.60 ± 77.94 ^b^	245.0 ± 16.0

The calories of the bamboo hemicellulose hydrolysates in the 5% BHH diets were calculated as 2 kcal/g (the equivalent of dietary fiber). Values are presented as mean ± SD. [*n* = 3 (Normal, *n* = 4 (Control and 5% BHH)]. Values in a column with different superscripts (a, b and c) indicate significant differences.

**Table 2 microorganisms-09-00888-t002:** Total cholesterol, triglyceride, and MDA levels in mice fed normal diets (Normal), high-fat diets (Control), or 5% BHH-containing high-fat diets (5% BHH) for 5 weeks.

Diets	Total Cholesterol (mg/mL)	Triglyceride (mg/mL)	MDA (µM)
Normal	69.19 ± 2.68 ^a^	56.53 ± 6.11	28.48 ± 4.49 ^a^
Control	129.96 ± 12.40 ^b^	62.70 ± 4.12	36.32 ± 4.83 ^b^
5% BHH	102.04 ± 17.52 ^c^	59.70 ± 21.56	34.08 ± 7.65 ^b^

Values are presented as mean ± SD [*n* = 3 (Normal), *n* = 4 (Control and 5% BHH)]. Values in a column with different superscripts (a, b and c) indicate significant differences.

## Data Availability

The data presented in this study are available on request from the corresponding author.

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
