# Peer review of "Fecal Microbiota Perspective for Evaluation of Prebiotic Potential of Bamboo Hemicellulose Hydrolysate in Mice: A Preliminary Study"

_microorganisms, 2021, doi:10.3390/microorganisms9050888_

Round 1

Reviewer 1 Report

Line 84-85 – the sum of moisture, crude protein, crude fat, crude fibre, crude ash and nitrogen free extract is 100.1%. It have to be 100.0%.  

Line 385 - … that that … (typing error).

The main shortage of this article is (as you wrote at the end of this article): the number of animals per group is too low. Therefore the scientific weight of results is questionable. Even though, for preliminary study can be this shortage acceptable. But in this case, I suggest to shown this information also at the end of title of this article (… - a preliminary study).

Reviewer 2 Report

This paper carefully treats the effect of prebiotic potential with Bamboo hemicellulose hydrolysate using a mice model, which affects the short-chain fatty acid production.

This is an interesting observation that may be a dietary potential for improving hypercholesteremia by modulating the gut microbial communities.
